# Surface temperature response to regional Black Carbon emissions: Do location and magnitude matter?

Maria Sand[1], Terje K. Berntsen[1,2], Annica Ekman[3,4], Hans-Christen Hansson[4,5], and Anna Lewinschal[3,4]

[1]CICERO Center for International Climate Research, Oslo, Norway
[2]Department of Geosciences, University of Oslo, Oslo, Norway
[3]Department of Meteorology, Stockholm University, Stockholm, Sweden
[4]Bolin Centre for Climate Research, Stockholm University, Stockholm, Sweden
[5]Department of Environmental Science and Analytical Chemistry, Stockholm University, Stockholm, Sweden

*Correspondence to*: Maria Sand (maria.sand@cicero.oslo.no)

**Abstract.** Aerosol radiative forcing can influence climate both locally and far outside the emission region. Here we investigate Black Carbon (BC) aerosols emitted in four major emissions areas and evaluate the importance of emission location and magnitude, as well as the concept of the absolute regional temperature-change potentials (ARTP). We perform simulations with a climate model (NorESM) with a fully coupled ocean and with fixed sea surface temperatures. BC emissions for year 2000 are increased by a factor of 10 and 20 in South Asia, North America and Europe, respectively, and by 5 and 10 in East Asia (due to higher emissions there). The perturbed simulations and a reference simulation are run for 100 years with 3 ensemble members each. We find strikingly similar regional surface temperature responses and geographical patterns per unit BC emission in Europe and North America, but somewhat lower temperature sensitivities for East Asian emissions. BC emitted in South Asia shows a different geographical pattern in surface temperatures, by changing the Indian monsoon and cooling the surface. We find that the ARTP approach rather accurately reproduces the fully-coupled temperature response of NorESM. Choosing the highest emission rate results in lower surface temperature change per emission unit compared to the lowest rate, but the difference is generally not statistically significant except for the Arctic. An advantage of high-perturbation simulations is the clearer emergence of regional signals. Our results show that the linearity of normalized temperature effects of BC is fairly well preserved despite the relatively large perturbations, but that regional temperature coefficients calculated from high perturbations may be a conservative estimate. Regardless of emission region, BC causes a northward shift of the ITCZ, and this shift is apparent both with fully coupled ocean and with fixed sea surface temperatures. For these regional BC emissions perturbations, we find that the effective radiative forcing is not a good measure of the climate response. A limitation of this study is the uncertainties on BC-cloud interactions and the amount of BC absorption, both which are model dependent.

## 1 Introduction

There has been a growing interest for reducing black carbon (BC) emissions to slow global warming and improve air quality (Jacobson, 2002; Shindell et al., 2012; Quinn et al., 2015; Sims et al., 2015). However, estimating the total climate impact

from BC is complicated because BC absorbs solar radiation and therefore rapidly influence heating rates, humidity and clouds in the atmosphere (Bond et al., 2013; Hansen et al., 1997;Cook and Highwood, 2004). These effects are often called 'rapid adjustments' and are distinct from the direct radiative forcing (DRF), i.e. scattering and absorption of sunlight by BC, and the indirect radiative forcing, i.e. changing the microphysical properties of clouds ((Twomey, 1977;Haywood and Boucher, 2000).

The surface warming by BC is highly sensitive to these rapid adjustments and is dependent on the altitude of the BC layer and the co-location of clouds (Ban-Weiss et al., 2012; Koch and Del Genio, 2010; Johnson et al., 2004; Jacobson, 2002). For instance in Johnson et al. (2004) using large-eddy simulations, it was found that BC within the boundary layer heated the air, decreased liquid water path (LWP) and thinned the marine stratocumulus clouds, while BC above the boundary layer increased LWP leading to a more shallow and moist boundary layer. Ramanathan et al. (2001) showed that BC can also reduce the

surface solar radiation and increase static stability lowering the surface moisture fluxes. Using prescribed BC in a general circulation model coupled to  a mixed-layer ocean Ban-Weiss et al. (2012) showed that BC near the surface caused surface warming and increased precipitation, while BC near the tropopause and in the stratosphere, on the other hand, decreased surface temperatures and decreased precipitation.  BC forcing from outside the region can contribute to the surface temperature response via transport of heat (Shindell, 2007; Sand et al., 2013; Menon et al., 2002).

Coupled Earth System Models (ESMs) now include most of the relevant processes to study the complex feedbacks mentioned above and could in principle be used to analyse and compare the effects of different regional/sectoral BC mitigation options of interest to policy makers. However, the change in emissions for these mitigation options are often too small (e.g. agricultural waste in Europe) to get a statistically significant signal in an ESM without running it for thousands of years, which makes it nearly impossible with state-of-the-art supercomputer clusters. As an alternative, climate metrics provide an easy way to

compare emission perturbations, forcing and response. The absolute regional temperature-change potentials (ARTPs) derived by Shindell and Faluvegi (2009) provide a relationship between forcing in one region and a surface temperature response in another region. The relation between forcing and surface temperature response were calculated for four different latitude bands (the Arctic, mid-latitudes, tropics and southern hemisphere (SH)), using a fully coupled climate model (GISS ModelE). Shindell and Faluvegi (2009) highlighted the importance of remote forcing on certain regions and found that the Arctic and

mid-latitudes were especially sensitive to the location of the forcing. For instance, the radiative forcing of BC at mid latitudes strongly influenced the surface temperature in the Arctic. In a follow-up-study Shindell et al. (2010) evaluated the method using transient historical simulations in four climate models and found good agreement. The ARTP coefficients have been used in many studies to estimate the surface temperature response to different forcing perturbations derived from chemistry transport models (Sand et al., 2015a; Collins et al., 2013). For instance, Sand et al. (2015a) estimated the Arctic surface

temperature response to emissions of BC, $SO_2$ and OC from a wide range of sectors and regions by calculating the direct radiative forcing (DRF) in four CTMs and applying the ARTP coefficients.

The ARTP method is quick and efficient, however, there are important simplifications underlying the calculations. As these coefficients cannot easily be validated by observations and have not been calculated in a multi-model experiment, there are

considerable uncertainties in these coefficients. Also, the emissions rates are calculated for broad latitude bands and any
longitudinal variations in the response is not represented.

Lewinschal et al. (2019) evaluated the surface temperature change in response to $SO_2$ emission perturbations in four major
emission regions using a different climate model (NorESM) and compared it to the temperature response estimated by using
the ARTPs. The ARTP method predicted similar latitudinal temperature response with those estimated by NorESM. Here we
perform simulations of the response to regional BC-emissions with the fully coupled climate model (NorESM) following the
setup used by Lewinschal for $SO_2$ emissions. We investigate the importance of emission location and magnitude and test the
ARTP concept. We calculate emission-to-temperature responses by estimating the regional surface temperature response to
BC emissions from four major emission regions; North America, Europe, South Asia, and East Asia. To get a signal of a small
emission perturbation in a coupled (atmosphere-ocean) climate model, it is often necessary to scale up the emissions. We
evaluate this assumed linearity by perturbing the BC emissions with two different emission rates.

## 2 Methods

### 2.1 NorESM

We have used the Norwegian Earth System model (NorESM1) (Bentsen et al., 2013;Iversen et al., 2013), which is largely
based on the CCSM4.0 framework (Gent et al., 2011) with special features for aerosols and their interaction with radiation and
warm cloud microphysics. (Kirkevåg et al., 2013;Seland et al., 2008). The model uses the finite volume dynamical core for
transport, with a horizontal resolution of 1.9° latitude by 2.5° longitude and 26 levels in the vertical. Unlike CCSM4.0,
NorESM1 is run with an elaborated version of the Miami Isopycnic Community Ocean Model (MICOM).

The aerosol life cycle scheme calculates mass concentrations of $SO_2$, BC, organic matter, sea salt, and mineral dust in up to
four size modes (nucleation, Aitken, accumulation, and coarse mode). BC is emitted in the nucleation, Aitken and accumulation
mode and in the internally mixed Aitken mode with organic matter. BC primary particles are assumed externally-mixed. BC
is internally-mixed with organic matter and $SO_4$, the latter through either condensation or coagulation. The mass of internally
mixed coating species is used in the calculations of the optical properties and cloud droplet number concentration, but it does
not alter wet deposition rates. Water is mixed into the aerosols based on the hygroscopicity and ambient relative humidity. The
calculated gas-phase components are DMS and $SO_2$. The aerosol concentrations are tagged according to different processes.
The processes include gas phase and aqueous phase chemical production, gas to particle nucleation, condensation on pre-
existing aerosol surfaces, and coagulation of smaller particles onto pre-existing Aitken, accumulation and coarse mode
particles. The process-tagged aerosol mass concentrations and relative humidity are given as input to look-up tables that
estimate the optical and physical properties of the aerosols. The aerosols can also act as cloud condensation nuclei (CCN)
based on their size and composition. BC that is mixed with other components and has become hydroscopic can contribute to
the number of CCN. Cloud droplet number concentration (CDNC) and liquid water path are prognostically calculated. CCN
activation is estimated based on supersaturations calculated from Köhler theory (Abdul-Razzak and Ghan, 2000) using a sub-

grid scale vertical velocity. The rate of activation depends on the size distribution and the hygroscopicity of the particles. Any cloud 'burn-off'-effects due to absorption by BC within individual cloud droplets are not included. Deposited aerosols on snow-cover and bare sea-ice are calculated prognostically and include hydrophobic and hydrophilic black carbon and dust. While the effects of deposition of these light-absorbing aerosols are taken into account, we have not calculated the radiative transfer of BC in snow explicitly-. A comprehensive assessment of the mean model state is available in Bentsen et al. (2013). They have shown that the global mean energy fluxes and associated cloud forcing are close to or within the observations range, but that the cloud cover is underestimated. The aerosols fields are evaluated in Kirkevåg et al. (2013). Here it is shown that global BC concentrations is underestimated by -18 % compared to observations. The global anthropogenic aerosol DRF is -0.08 Wm$^{-2}$ and the indirect radiative forcing is -1.2 Wm$^{-2}$.

## 2.2 Experimental Set-up

Coupled atmosphere/ocean simulations with NorESM have been performed for a set of emission perturbations. BC emissions have been increased in four areas: South Asia, East Asia, North America and Europe. In these regions the emissions have been perturbed by factors of 10 and 20, except East Asia that has been perturbed by 5 and 10 (due to the relative higher emissions in this region). The magnitude of these rates was chosen to get a large enough signal, and at the same time keep global DRF below ~1 W/m$^2$. Figure S1 shows the increase in total global emissions compared to a baseline simulation for each region. The baseline is run with year 2000 annually repeating emissions of aerosols and precursors, land use conditions and greenhouse gas concentrations. The aerosol emissions (and their precursors) are the historical emissions of CMIP5 described in Lamarque et al. (2010). The global mean BC emissions for the baseline simulation are 7.7 Tg/yr. Table 1 provides an overview of all the simulations. For each perturbation coupled simulations for 100 years with 3 ensemble members have been run, i.e. a total of 300 years of simulation per emission perturbation. Furthermore, we calculate the DRF as the instantaneous change in energy flux at the top-of-the-atmosphere with a double-call in the radiation routine. The calculations of the DRF does not include the BC-on-snow-forcing. In separate simulations with prescribed sea surface temperatures and sea-ice cover we calculate the effective radiative forcing (ERF) which include rapid tropospheric adjustments, often related to humidity and clouds that may exhibit fast adjustments to the radiative forcing and any changes in CCN. We have calculated the ERF as the annual mean change in the net radiation at TOA after the atmosphere is allowed to respond to the forcing (i.e. rapid adjustments in temperature, relative humidity and clouds), over the last 10 years of total 15 years simulations. The surface temperature response per unit of DRF, ERF and per emission change is then compared for the four emission areas. All numbers reported are annual means from the last 80 years of the simulations (i.e. 80×3 years for each perturbation experiment). 20-year spin-up was sufficient for equilibrium to occur for BC perturbations (this is also shown in Sand et al., 2015b)

## 3 Results

Figure 1 shows the surface temperature change for the perturbed emission runs compared to baseline. The broad patterns are similar to the well-known warming pattern due to CO2 forcing, with larger warming over land and in the northern areas. The geographical pattern of warming is comparable for BC emitted in North America, Europe and East Asia with only small longitudinal differences between those perturbations. For South Asia, however, there is a significant local cooling effect over India. This cooling is linked to a strong decrease in downwelling SW radiation over India caused by the absorption and scattering by BC aerosols, less upward motion and an increase in low level clouds. Northern hemisphere BC emissions cause a northward displacement of the ITCZ that is most pronounced for emissions in South Asia and a corresponding shift in the Indian monsoon. The tropospheric jet is moving northward to adjust for a northward shift in the meridional temperature gradient (not shown).This type of displacement caused by a hemispherically asymmetric heating is a robust feature found in several climate model studies (Chung and Seinfeld, 2005; Jones et al., 2007; Meehl et al., 2008; Allen et al., 2012; Voigt et al., 2017;Kovilakam and Mahajan, 2015).

Figure 2 shows the regional surface temperature change in four latitude bands (The Arctic (60°N-90°N, mid latitude (28°N-60°N), Tropics (28°S-28°N) and SH (90°S-28°S)) normalized by the enhancement in the BC emissions. Table 2 provides the corresponding numbers. For all cases the Arctic region shows the highest sensitivities, followed by the mid latitudes. The temperature sensitivities in the tropics and SH are lower compared to the mid latitudes and the Arctic, but these two regions are also much larger regions compared to the mid latitudes and the Arctic. These zonal mean temperature sensitivities are similar regardless of emission location, especially for BC emitted in North America and Europe. The temperature sensitivities are slightly lower in the Arctic and mid-latitudes for BC emitted in Asia, in particular South Asia. Temperatures in the Arctic increase by 0.04-0.1 K per Tg/yr BC emitted. In the mid latitudes the temperature increases by 0.04-0.08 K per Tg/yr BC emitted.

The two twin-bars in Fig. 2 given for each emission region represent the linearity test. The lighter coloured bars are the perturbations with the lowest emission rates, while the darker colour bars are the runs with the highest rates. In general, the sensitivity decreases with increasing emissions, most pronounced for the response in the Arctic (compare e.g. 20×EU vs. 10×EU, i.e. the right and left columns in Fig. 2 or numbers in Table 2). The decrease in Arctic sensitivities for the highest emissions are significant for BC emitted in Europe and Asia, but not North America ($p < 0.05$). For the mid latitude sensitivities, the changes are small, and only European emissions show significant changes (again, with weaker sensitivities for 20×emissions compared to 10×emissions).

To understand the regional differences and the non-linarites we first investigate the relation between emissions and concentrations. Figure 3 shows the change in regional BC burden normalized to global BC emissions. As expected, the burden changes per global emissions are largest in the latitude band where the emissions are located. The regional burden change normalized to emissions is slightly higher for the highest emission rates. These changes are statistically significant using a 0.05 level of significance (and also using a 0.005 level), except for the changes in the Tropics and SH for North American

emissions. The same results are also obtained when we look at total global burden change (not shown). E.g. in Europe, emitting 20× emissions results in a 117 % increase in global BC burden compared to 10×emissions. The rapid adjustments of BC cause a local warming of the atmosphere and an increased local convection pushing the BC layers higher up in the atmosphere where the BC is less likely to be scavenged. For the simulations with the high emission rate, the relative concentration increase normalized to global emissions is largest at high altitudes above 200 hPa (25-55% increase shown in zonal mean plots of concentrations normalized to global emissions per altitude in Fig S3-S6). This effect has also been shown in Sand et al. (2015b) for BC perturbed by increasing the BC emission rate compared to increasing the BC concentrations. As in this study, increasing the emission rate resulted in a relative increase in BC at higher altitudes compared to lower altitudes and a longer BC residence time. This effect may explain why the temperature increase per emissions is largest for the small emission rate. Although BC perturbations at higher altitudes cause a larger DRF per unit burden change (Samset and Myhre, 2011), this is out weighted by a smaller temperature response per unit forcing. As BC is transported higher up in the atmosphere the surface temperature response decrease, as shown in Ban-Weiss et al. (2012). In the Arctic the effect of high-altitude BC is particularly strong, even causing a cooling at the surface for positive regional DRF, because of the strong vertical stability in this area (Flanner, 2013; Sand et al., 2013). A similar weaker temperature efficiency with stronger emission perturbations have also been found in Yang et al. 2019, but for much higher emission perturbations.

Figure 4 a) shows the climate sensitivity in terms of regional surface air temperature change per global DRF, i.e. a similar concept as the ARTPs (but for global forcing). In general, the temperature sensitivities for BC emitted in Europe and North America are similar, and 50% lower for BC emitted in East Asia and South Asia. In the Arctic the temperature increases 0.8-1.7 K per $W/m^2$, while in the mid-latitudes the temperature increases by 1 K per $W/m^2$. The Arctic region shows a higher climate sensitivity for the lower emission rate compared to the higher emission rate (the difference is not significant for North American emissions). The ERF is defined as the net SW minus the net LW for fixed SSTs perturbations. The temperature sensitivities when using ERF instead for DRF (Fig 4b), do not show the same consistent pattern The rapid adjustments of BC are strongly negative in our model, and partly offset the DRF; resulting in lower ERF (-0.0004 - 0.2 $W/m^2$) compared to DRF (0.3 - 1.3 $W/m^2$). The small value of ERF makes the sensitivities ($K/Wm^{-2}$) large and uncertain (maps of ERF are shown in S02). For 20×South Asian emissions, the (global mean) ERF is close to 0 (which makes the climate sensitivity defined here undefinedly large). Even though ERF may not be a good predictor of the surface temperature response, only using DRF will hide some of the uncertainties related to atmospheric absorption. The rapid adjustments of BC cause a shift in the large-scale circulation patterns in the atmosphere. This shift is apparent both when using a coupled ocean and fixed SSTs. As an example, the net TOA SW and LW fluxes are plotted for 10×East Asia for both ocean setups in Fig 5. The ERF is positive close to (all) the emission perturbations, but shows negative values linked to the northward shift in the ITCZ.

The rapid adjustments to BC perturbations include changes in cloud cover. Figure 6 shows the global maps of changes in high cloud and low clouds. High cloud cover is slightly reduced in all runs and are surprisingly consistent between the runs. Low cloud cover increases globally for all simulations, especially for marine stratocumulus cloud regions. This increase of low cloud cover outside the west coast of North America is largest for the Asian emission increases and is correlated with surface

cooling. In the emission areas local low cloud cover decrease, except for South Asian emissions where local low clouds increase. As in most other models, BC is not included as an ice nuclei in NorESM. This might have an impact on the monsoon response and clouds. The version of NorESM we have used is known to have a relatively large convective transport of aerosols (Allen and Landuyt, 2014), which may cause an overestimation of BC in the upper troposphere.

Simulating the temperature response with coupled ESM requires large computer resources (here we have used 300 model year simulations for each experiment) due to the internal variability of the model. Alternatively, one can estimate the temperature response in broad latitude bands by using the ARTP approach. We compare our annual mean surface temperature estimates from the coupled simulations with estimates using the much simpler ARTP-based approach based on the regional response coefficients (in K/Wm$^{-2}$) from Shindell and Faluvegi (2009). This is done by calculating the temperature responses in each band by using their response coefficients multiplied with our forcing estimates (i.e. the DRF), see Collins et al., 2013. Please note that the coefficients from Shindell and Faluvegi (2009) are normalized by DRF, and the rapid adjustments in the atmosphere are thus included in the regional response coefficients (as calculated in the coupled runs by Shindell & Faluvegi (2009)). Figure 7 shows the comparison (temperature sensitivities in K/(Tg (BC)yr$^{-1}$) per latitude band) between the ARTP-based method and our coupled model estimates. For the ARTP calculations in the Arctic response region, we have used numbers from Sand et al (2015a), which includes vertically resolved forcing and sea-ice/snow forcing. This is not included in the other latitude bands (as we did not have it available). GISS ModelE and NorESM have the same equilibrium climate sensitivity (2.9 K). By reconstructing the temperature response, we obtain a fairly comparable response, even though the ARTP constructed response varies more between the emission regions in each latitude band. Reasons for this might be differences in the vertical distributions of BC and indirect effects.

Lewinschal et al. (2018) calculated the same emission-to-temperature responses using NorESM, but for SO$_2$ emissions. They found that the temperature response was independent of emission location and as a global average equal to -0.006 K/TgSyr$^{-1}$. As in this study, the Arctic was found to have the largest temperature response in all simulations. Here, we find global surface temperature responses per unit BC emitted to vary between the emission regions, with a systematic north-south gradient; 0.029 (or 0.021 with high emission rate) K/Tg yr$^{-1}$ for European emissions; 0.025 (0.023) K/Tg yr$^{-1}$ for North American emissions; 0.019 (0.016) K/Tg yr$^{-1}$ for East Asian emissions; and 0.017 (0.015) K/Tg yr$^{-1}$ for South Asian emissions.

## 4 Summary and conclusion

We have estimated the temperature responses for BC emitted in four major emission regions in the Northern Hemisphere. The largest temperature response was found in the Arctic, independent of emission region. Generally, the response is similar whether BC is emitted in North America or Europe, and to some extent also East Asia (but with slightly lower response in the Arctic). For BC emitted in South Asia the response is weaker due to a strong decrease in downwelling solar radiation and local surface cooling due to a displacement of the Indian monsoon. Regardless of emission region, BC cause a northward shift in the ITCZ. This is apparent both when using a coupled ocean and with fixed SSTs.

The Arctic temperature change per unit emissions depends on the magnitude of the forcing. The higher the emission rate, the lower the temperature sensitivity. This non-linearity is partly because enhanced absorption in the highest emission cases increase vertical mixing so that of BC is transported higher up in the atmosphere, which decreases the surface temperature response, a feature also shown in (Ban-Weiss et al., 2012). For considerably lower emissions, the sensitivity could be higher. This result implies that the regional temperature coefficients calculated from high emission perturbations may be a conservative estimate, but in general that the linearity of normalized temperature effects of BC is fairly well preserved in our model.

When comparing our global temperature responses per unit BC emitted to the response to $SO_2$ emissions for the same regions also using the NorESM model (Lewinschal et al., 2019), we find that the BC sensitivities are 3-5 times larger compared to $SO_2$. The global temperature responses to BC also vary between the emission regions with a systematic north-south gradient. By reconstructing the temperature response using the much simpler ARTP method, we find that the ARTP method works quite well, but that there are regional differences within the latitude bands, especially linked to circulation changes and the Indian monsoon.

### Data availability

The model data can be made available upon request by the corresponding author. The source code for NorESM is available at https://github.com/metno/noresm.

### Author contribution

MS performed the simulations, figures/statistics, and lead the writing of the paper. All authors contributed to the design of the study, the analysis of the results and the writing of the paper.

### Competing interests

The authors declare that they have no conflict of interest.

### Acknowledgements

This work has been supported by the Research Council of Norway through the projects EVA (229771), AC/BC (240372), and BlackArc (240921). The simulations were performed on resources provided by UNINETT Sigma2 – the National Infrastructure for High Performance Computing and Data Storage in Norway (nn9188k).

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

## Surface temperature change [K]

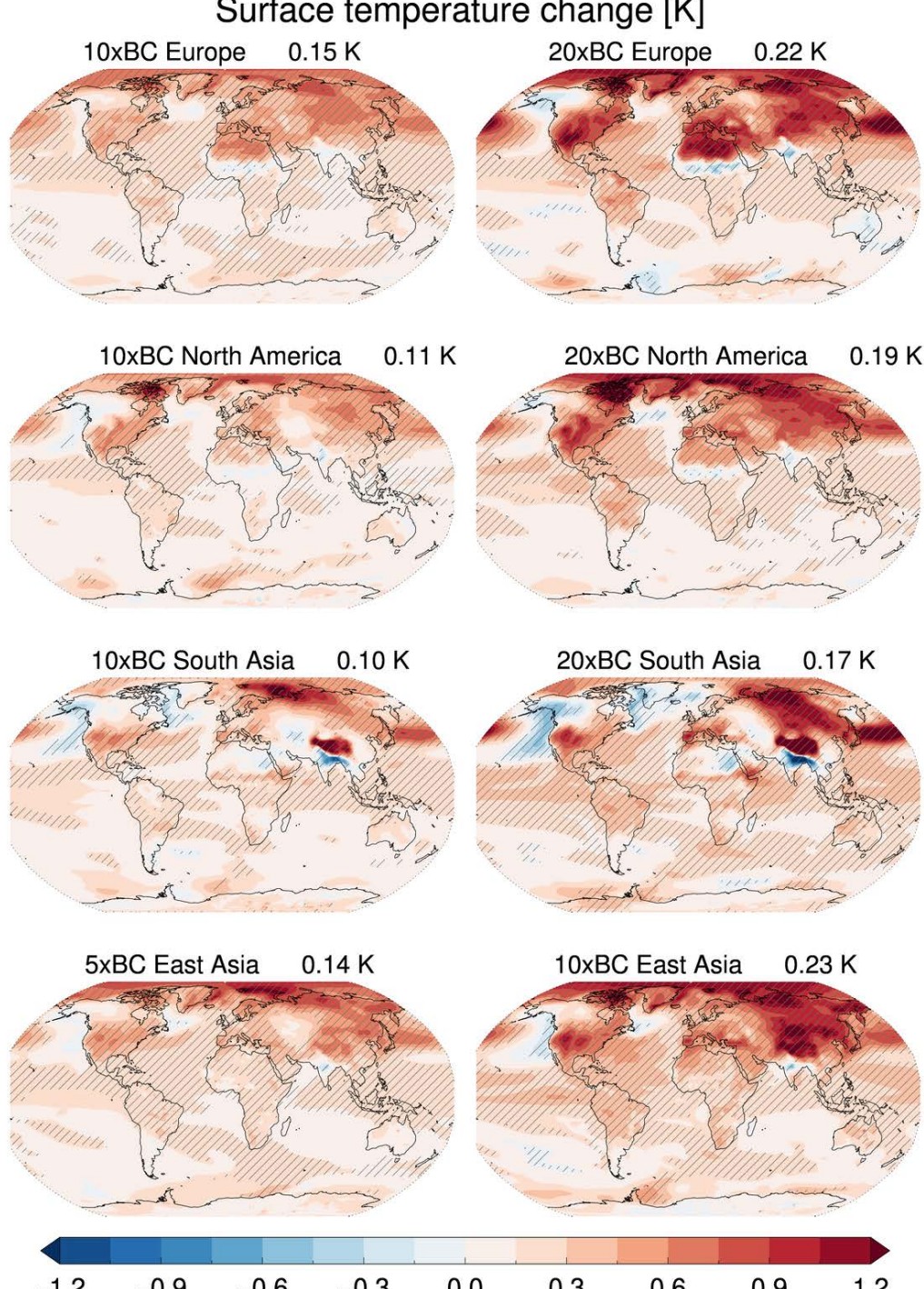

**Figure 1: Surface temperature change (in K) for the perturbed runs minus the baseline. The emission region is given on each panel (from top: Europe, North America, South Asia, East Asia). In the right-side column the emission rate is doubled compared to the left-side column. The stippled areas represent statistically significant changes (p<0.05).**


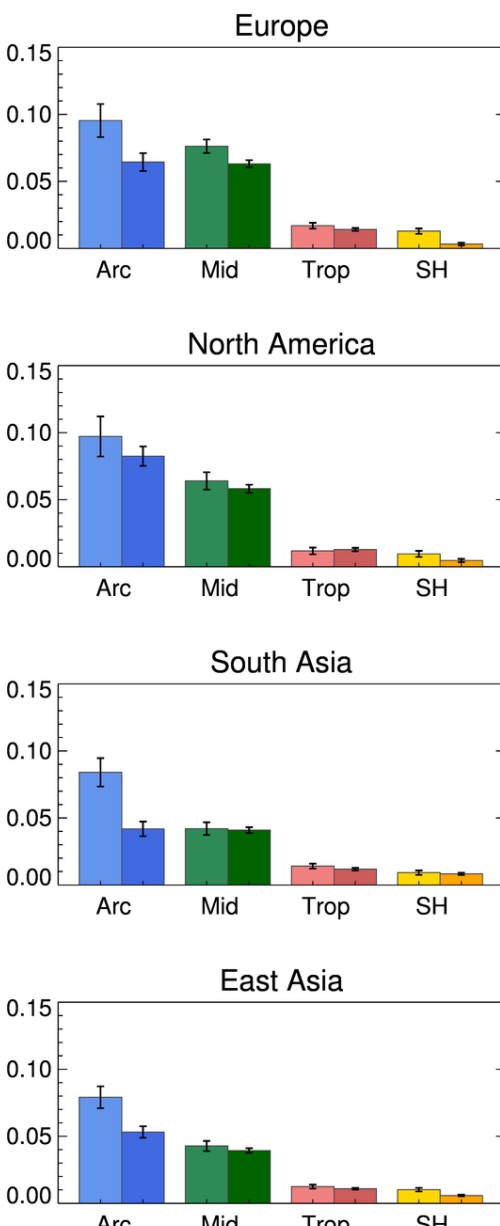

**Figure 2: Regional surface temperature response per BC emissions (K/Tgyr$^{-1}$). The emission location (Europe, North America, South Asia, East Asia) is given on top of each plot. The response region (latitude band) is given on the x-axis. The emission rate is doubled on the right-side bars (darker colours) compared to the left-side bars (lighter colours). The error bars represent the standard error of mean.**

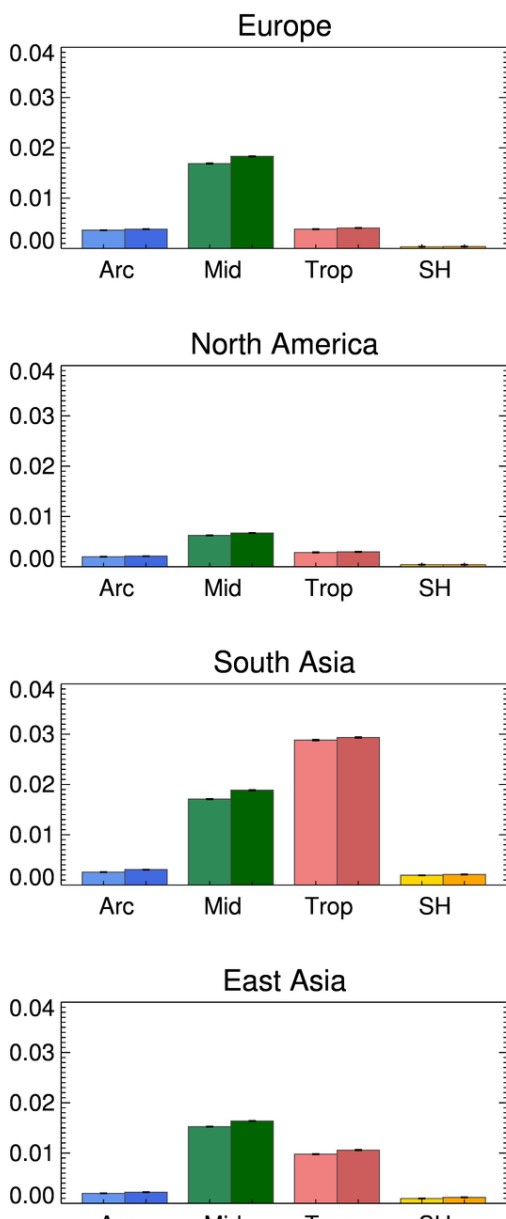

**Figure 3: Regional BC burden change per global BC emissions (Tg/Tgyr$^{-1}$). The emission location (Europe, North America, South Asia, East Asia) is given on top of each plot. The response region (latitude band) is given on the x-axis. The emission rate is doubled on the right-side bars (darker colours) compared to the left-side bars (lighter colours). The error bars represent the standard error of mean.**


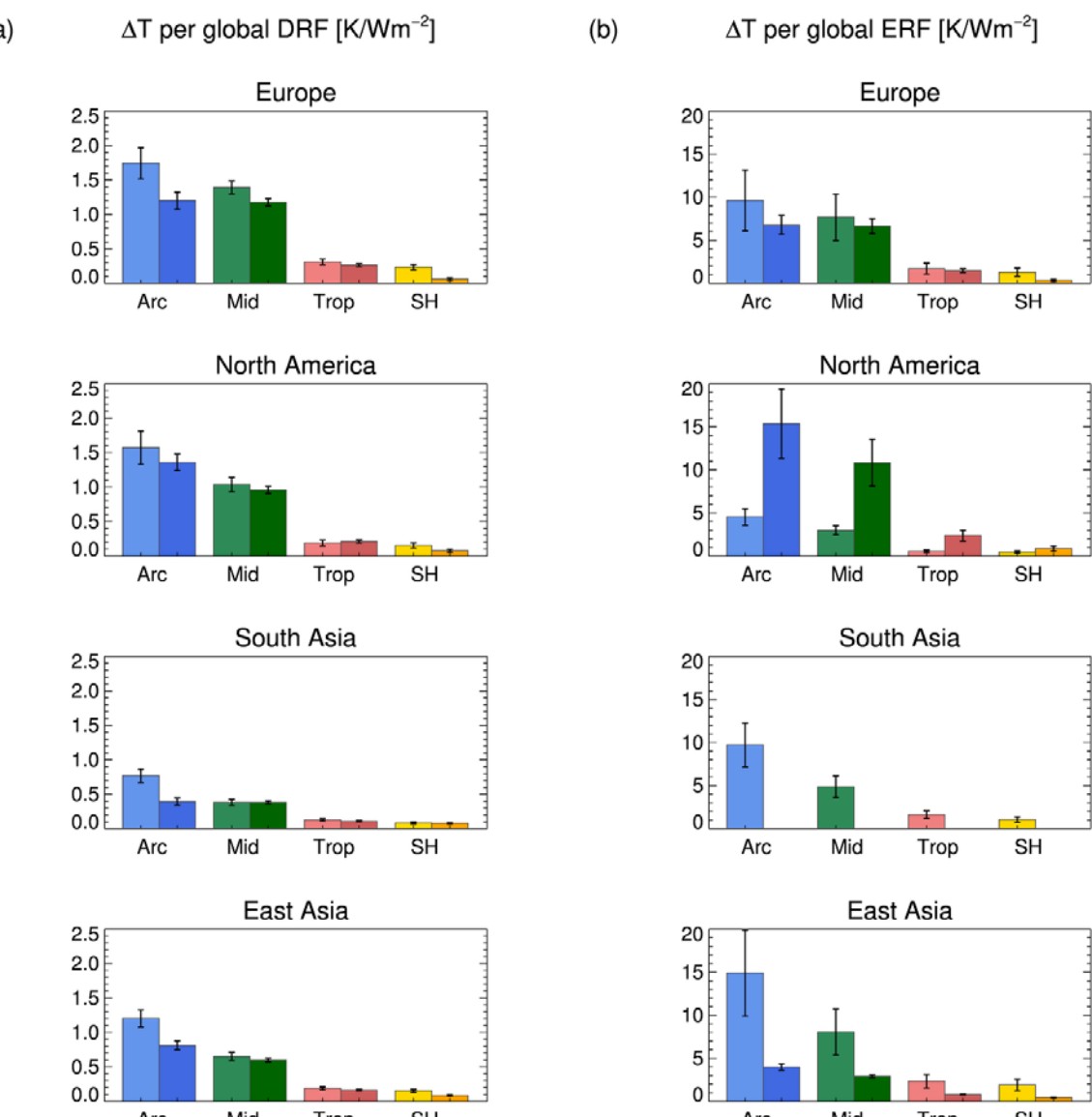

**Figure 4: Regional surface temperature response per global BC (a) DRF and (b) ERF (in K/Wm⁻²). The emission location (Europe, North America, South Asia, East Asia) is given on top of each plot. The response region (latitude band) is given on the x-axis. The emission rate is doubled on the right-side bars (darker colours) compared to the left-side bars (lighter colours). The error bars represent the standard error of mean.**

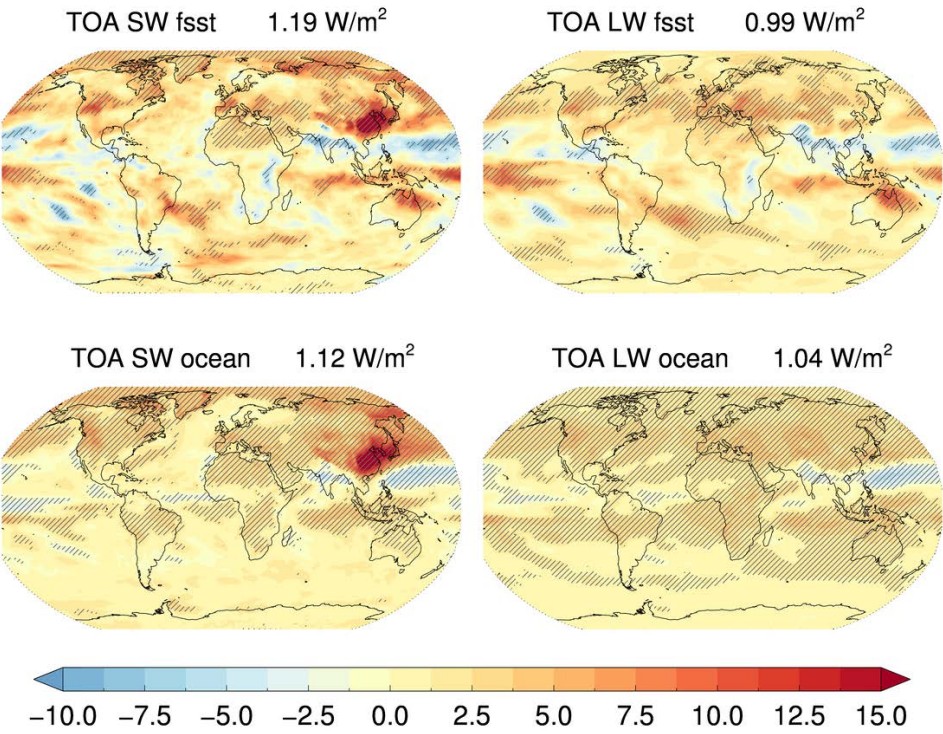

**Figure 5: Change in delta TOA net SW and LW radiative fluxes (sum of downward and upward) for the 10×BC in East Asia perturbation minus the baseline. Top row shows the runs with fixed SSTs, and bottom row shows the runs with fully coupled ocean. The stippled areas represent statistically significant changes (p<0.05).**

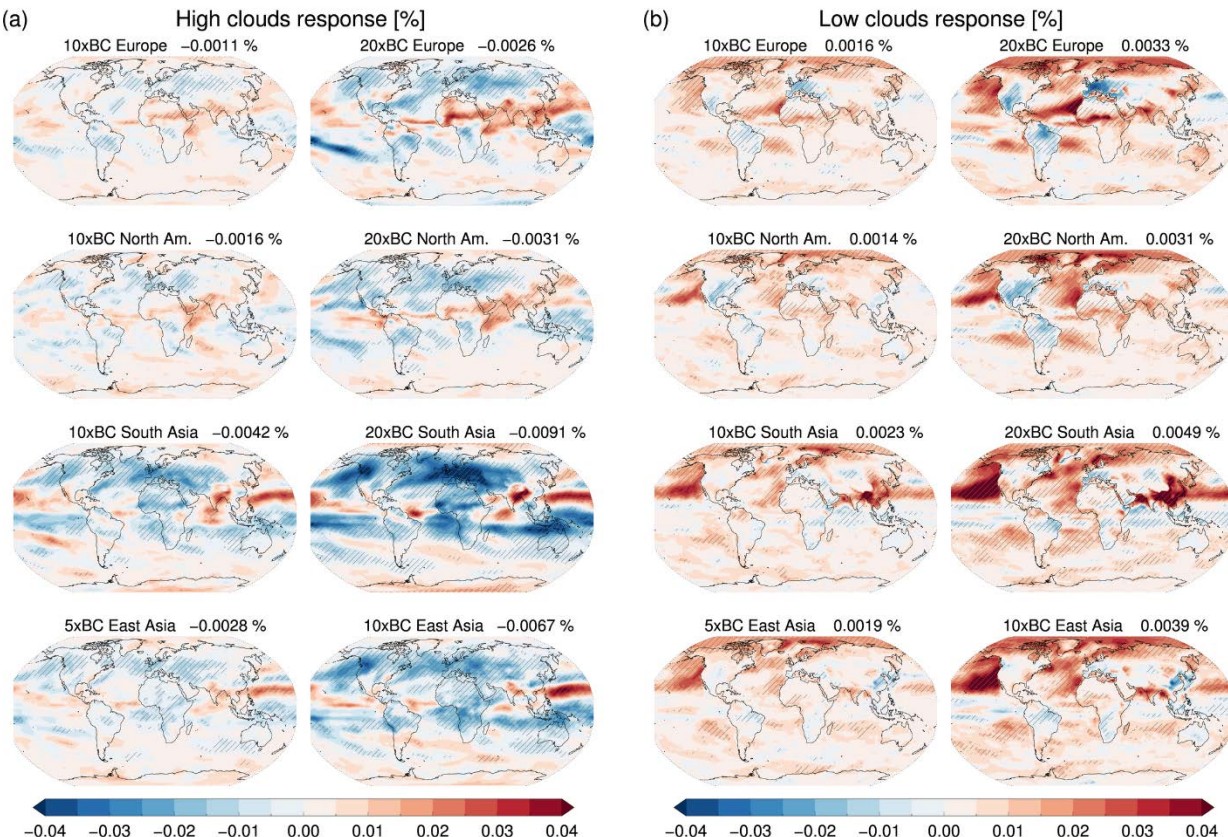


**Figure 6: Response in high clouds (located above 400 hPa) (a) and low clouds (surface up to 700 hPa) (b) in (%). The emission location (Europe, North America, South Asia, East Asia) is given on top of each plot. In the right-side column the emission rate is doubled compared to the left-side column.**

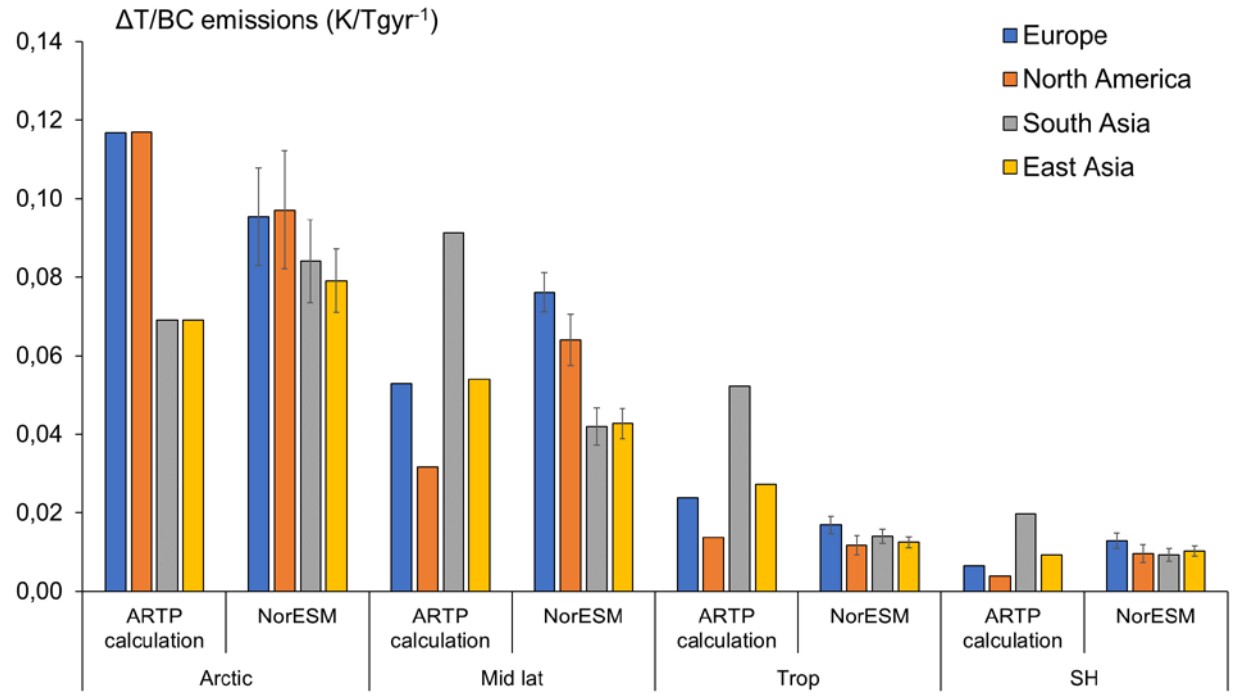


**Figure 7: A comparison between ARTP calculated and NorESM calculated regional surface temperature response per unit BC emissions (K/Tg/yr⁻¹). The lowest emission perturbation is shown here (10×Europe, 10×North America, 10×South Asia, 5×East Asia). The response region (latitude band) is given on the x-axis. The error bars for NorESM represent the standard error of mean.**


## Tables

**Table 1: Overview of the model simulations in this study and the global BC emissions in each simulation (in Tg BC/yr).**

| | | |
|---|---|---|
| Control | Year 2000 control run 3×100 years fully coupled equilibrium simulations | 7.7 |
| 10 × Europe | Same as control, but with BC emissions in Europe multiplied by 10 | 12,8 |
| 20 × Europe | Same as control, but with BC emissions in Europe multiplied by 20 | 17,9 |
| 10 × North America | Same as control, but with BC emissions in North America multiplied by 10 | 11,9 |
| 20 × North America | Same as control, but with BC emissions in North America multiplied by 20 | 16,1 |
| 10 × South Asia | Same as control, but with BC emissions in South Asia multiplied by 10 | 13,5 |
| 20 × South Asia | Same as control, but with BC emissions in South Asia multiplied by 20 | 19,3 |
| 5 × East Asia | Same as control, but with BC emissions in East Asia multiplied by 5 | 15,0 |
| 10 × East Asia | Same as control, but with BC emissions in East Asia multiplied by 10 | 22,4 |

**Table 2: Surface temperature change per global BC emissions (in K/Tg yr$^{-1}$). Bold numbers represent statistically significant differences between the emission rates (using 10 vs. 20, or 5 vs. 10 for East Asia).**

|  | **Arctic** | **Mid lats** | **Tropics** | **SH** |
|---|---|---|---|---|
| 10 × Europe | **0,10** | **0,08** | 0,02 | **0,01** |
| 20 × Europe | **0,06** | **0,06** | 0,01 | **0,00** |
| 10 × North America | 0,10 | 0,06 | 0,01 | 0,01 |
| 20 × North America | 0,08 | 0,06 | 0,01 | 0,00 |
| 10 × South Asia | **0,08** | 0,04 | 0,01 | 0,01 |
| 20 × South Asia | **0,04** | 0,04 | 0,01 | 0,01 |
| 5 × East Asia | **0,08** | 0,04 | 0,01 | **0,01** |
| 10 × East Asia | **0,05** | 0,04 | 0,01 | **0,01** |

**Table 3: Surface temperature change per global direct radiative forcing (in K/Wm$^{-2}$). Bold numbers represent statistically significant differences between the emission rates (using 10 vs. 20, or 5 vs. 10 for East Asia).**

|  | **Arctic** | **Mid lats** | **Tropics** | **SH** |
|---|---|---|---|---|
| 10 × Europe | **1,74** | **1,39** | 0,31 | **0,23** |
| 20 × Europe | **1,20** | **1,18** | 0,26 | **0,06** |
| 10 × North America | 1,57 | 1,04 | 0,19 | 0,15 |
| 20 × North America | 1,36 | 0,96 | 0,21 | 0,08 |
| 10 × South Asia | **0,77** | 0,38 | 0,13 | 0,08 |
| 20 × South Asia | **0,39** | 0,39 | 0,11 | 0,08 |
| 5 × East Asia | **1,20** | 0,65 | 0,19 | **0,15** |
| 10 × East Asia | **0,81** | 0,60 | 0,17 | **0,09** |
