# Peer review of "Surface temperature response to regional Black Carbon emissions: Do location and magnitude matter?"

_Atmospheric Chemistry and Physics, 2019_

## Referee Comment (RC1) · Anonymous Referee #2 · 3 Oct 2019

This study explores the dependency of emissions-normalized radiative forcing and regional temperature response to the location and magnitude of black carbon emissions, using numerous simulations with the NorESM model. Given the struggles of past efforts to reliably quantify climate impacts from small emissions of short-lived species, this study provides very useful information that can be applied to inform future efforts, such as those of the AMAP expert group on short-lived climate forcers. I recommend publication of this article in ACP after the following minor issue are addressed.

Minor issues:

Abstract: I suggest including a statement on your findings of the ARTP approach, or

namely that this approach (when used with DRF) rather accurately reproduces the fully-coupled temperature response of NorESM.

It is a bit surprising to me that the ITCZ can shift in fixed-SST simulations. If you are aware of other studies that have shown this behavior (especially any of the studies cited on lines 104-105), I suggest noting this.

It is mentioned that the BC-on-snow effect is included in NorESM, but is it included in the estimates of DRF? Please clarify and comment on any associated implications.

Section 2.2: More details are needed on the experimental set-up. Specifically: Were annually-repeating or annually-changing boundary conditions (GHGs, SSTs, etc) used in the runs, and from which years do the conditions represent? How long were the ERF fixed-SST runs and what time frame was used for the analysis? (line 93: "after the atmosphere is allowed to respond to the forcing" needs precision). The text indicates that the final 80 years of 100-year fully-coupled runs were analyzed. Is 20 years sufficient for equilibration to occur? Perhaps for BC it is, but a brief discussion on this should be added.

Please describe the baseline emissions that are used, e.g., which inventory are they from and what are the global BC emissions? I also suggest adding the annual emissions for each experiment/region to Table 1.

line 189: Sentence beginning "For South Asia..." - Does this South Asia response occur only with emissions from South Asia, or is it also seen with emissions from elsewhere that transport to South Asia?

Technical issues:

line 14: "a rate of" -> "a factor of"

line 17: "BC emitted in South Asia shows a different geographical pattern by changing..." - This sentence needs clarification. Geographical pattern of which variable(s)?

[Figure]

line 75: "according different" -> "according to different"

line 84: "perturbed by 10 and 20" -> "perturbed by factors of 10 and 20"

line 128: "global BC burden": This is the  global BC burden, correct?

line 132: Please clarify the meaning of "emission-driven vs. concentration-driven BC"

line 144-145: Sentence beginning "For the Arctic..." is unclear. Please rework this.
* * *

---

## Referee Comment (RC2) · Anonymous Referee #3 · 19 Nov 2019

Review of "Surface temperature response to regional black carbon emissions: Do location and magnitude matter?" by Sand et al., submitted to Atmospheric Chemistry and Physics.

This paper discusses the impacts of regional emissions of black carbon on global and regional temperatures. Scenarios are examined with different levels of emissions. One conclusion is that a higher emission rate results in lower temperature change per unit emissions, except in the Arctic. Impacts of emissions from North America and Europe are found to be similar to each other but different from emissions from East Asia.

Overall, the paper contributes to our understanding of the impacts of black carbon.

[Figure]

However, the paper can be improved if the model is described more completely and if previous understanding of black carbon climate effects were discussed more completely. Also, the authors do not discuss the physical processes they are not including that could affect results. Finally, there is no evaluation of the model. Below are more specific comments.

With respect to the model, there should be some validation of baseline cloud fields and aerosol optical depth and aerosol absorption optical depth against satellite and/or in situ data. Some validation is required in virtually all papers.

Abstract. "BC emissions are increase by a rate of 10 and 20..." 10 and 20 what? A factor of 10 and 20? 10 and 20%? Please clarify.

Introduction. "There has been a growing interest for reducing black carbon emissions to slow global warming emissions and improve air quality." This was proposed at least as far back as 2002 in Jacobson (2002), which states in the abstract, "Reducing BC + OM will not only slow global warming but also improve human health." Please include this information.

Introduction. "BC absorbs solar radiation and therefore rapidly influence heating rates, humidity and clouds in the atmosphere." That feedback, along with 11 other feedbacks of black carbon to climate, is discussed Jacobson (2002), Sections 3.1-3.12. Please discuss.

NorESM. Please state explicitly how, in your model, "aerosols can act as CCN based on their size and composition." Is it an empirical parameterization? Do cloud drops of different size and composition grow physically in time, or do you assume an equilibrium cloud drop size distribution.

Please state explicitly in the text whether you include Cloud Absorption Effects I and II of black carbon (Jacobson, 2012). CAE I is cloud burnoff due to absorption by black carbon inclusions within individual cloud drops and CAE II is cloud burnoff due the

hydration of BC-containing aerosol particles between cloud drop at the ~100% relative humidity of the cloud, and the resulting enhanced absorption and heating of the cloud due to optical focusing. If you do not include it, please discuss briefly the potential impacts of not including this treatment.

Please state explicitly how "The BC on snow effect is included in the model." Do you model the size-resolved deposition of BC (through wet and dry deposition) to the surface. How is radiative transfer solved through the snow if it is? If it is not, please state so explicitly.

What is the mixing state of BC aerosol components in the model? Externally mixed, well-internally mixed, or something in between?

Abstract. "For these regional BC emissions perturbations, we find that the effective radiative forcing is not a good measure of the climate response." Although it is not clear if your model includes this, one explanation may be that "when absorbing aerosols exist in clouds, instantaneous direct radiative forcing (DRF) and surface temperature change are anticorrelated because when absorbing aerosol burns off a cloud, the aerosol DRF decreases due to a decrease in optical focusing, yet surface temperature escalates rapidly due to the pouring in of sunlight to the surface." (Jacobson, 2014). Please discuss.

Figure 5. Change figure captions to "Delta TOA" rather than just "TOA" Are these "net downward minus upward" irradiances with versus without black carbon emissions? If so, please clarify.

The text mentions that the Indian Monsoon was displaced. How was it displaced? In what direction and what was the magnitude in wind speed change? Were Monsoon wind speeds decreased due to black carbon? Were globally or regionally averaged wind speeds decreased as might be expected (Jacobson and Kaufman, 2006).

References

Jacobson, M. Z., Control of fossil-fuel particulate black carbon plus organic matter, possibly the most effective method of slowing global warming, J. Geophys. Res., 107 (D19), 4410, doi:10.1029/ 2001JD001376, 2002

Jacobson, M.Z., and Y.J. Kaufmann, Wind reduction by aerosol particles, Geophys. Res. Lett., 33, L24814, doi:10.1029/2006GL027838, 2006.

Jacobson, M.Z., Investigating cloud absorption effects: Global absorption properties of black carbon, tar balls, and soil dust in clouds and aerosols, J. Geophys. Res., 117, D06205, doi:10.1029/2011JD017218, 2012.

Jacobson, M.Z., Effects of biomass burning on climate, accounting for heat and moisture fluxes, black and brown carbon, and cloud absorption effects, J. Geophys. Res., 119, 8980-9002, doi:10.1002/2014JD021861, 2014.
* * *

---

## Author Comment (AC1) · 2 Dec 2019

*We would like to thank the reviewer for the dedicated time he/she spent on reading the manuscript and providing valuable feedback to us. A point-by-point response is given below.*

This study explores the dependency of emissions-normalized radiative forcing and regional temperature response to the location and magnitude of black carbon emissions, using numerous simulations with the NorESM model. Given the struggles of past efforts to reliably quantify climate impacts from small emissions of short-lived species, this study provides very useful information that can be applied to inform future efforts, such as those of the AMAP expert group on short-lived climate forcers. I recommend publication of this article in ACP after the following minor issue are addressed.

Minor issues:

Abstract: I suggest including a statement on your findings of the ARTP approach, or namely that this approach (when used with DRF) rather accurately reproduces the fully-coupled temperature response of NorESM.

*Thank you for pointing this out. We have added such a statement to the abstract.*

It is a bit surprising to me that the ITCZ can shift in fixed-SST simulations. If you are aware of other studies that have shown this behavior (especially any of the studies cited on lines 104-105), I suggest noting this.

*Yes, this is part of the fast response resulting from atmospheric heating and land cooling before sea-surface temperature (SST) has time to respond and this causes a northward ITCZ shift. This has been shown in Voigt et al. (2017) for seven atmosphere models with large aerosol forcing. We have added this reference.*

*Voigt, A., Pincus, R., Stevens, B., Bony, S., Boucher, O., Bellouin, N., Lewinschal, A., Medeiros, B., Wang, Z., and Zhang, H. ( 2017), Fast and slow shifts of the zonal-mean intertropical convergence zone in response to an idealized anthropogenic aerosol, J. Adv. Model. Earth Syst., 9, 870– 892, doi:10.1002/2016MS000902.*

It is mentioned that the BC-on-snow effect is included in NorESM, but is it included in the estimates of DRF? Please clarify and comment on any associated implications.

*We have clarified that the DRF only include the atmospheric forcing, and not BC-on-snow. The BC-on-snow effect is included when calculating the surface temperature response/albedo changes etc.*

Section 2.2: More details are needed on the experimental set-up. Specifically: Were annually-repeating or annually-changing boundary conditions (GHGs, SSTs, etc) used in the runs, and from which years do the conditions represent? How long were the ERF fixed-SST runs and what time frame was used for the analysis? (line 93: "after the atmosphere is allowed to respond to the forcing" needs precision). The text indicates that the final 80 years of 100-year fully-coupled runs were analyzed. Is 20 years sufficient for equilibrium to occur? Perhaps for BC it is, but a brief discussion on this should be added.

Please describe the baseline emissions that are used, e.g., which inventory are they from and what are the global BC emissions? I also suggest adding the annual emissions for each experiment/region to Table 1.

*The have added the following sentences in Methods to clarify:*

*'We have calculated the ERF as the annual mean change in the net radiation at TOA after the atmosphere is allowed to respond to the forcing (i.e. rapid adjustments in temperature, relative humidity and clouds), over the last 10 years of total 15 years simulations '*

*'The baseline is run with year 2000 annually repeating emissions of aerosols and precursors, land use conditions and greenhouse gas concentrations. The aerosol emissions (and their precursors) are the historical emissions of CMIP5 described in Lamarque et al. (2010). The global mean BC emissions for the baseline simulation are 7.7 Tg/yr.'*

*'20 year spin-up was sufficient for equilibrium to occur for BC perturbations (this is also shown in Sand et al, (2015))'*

*We have also added a column with global annual BC emissions in Table 1.*

line 189: Sentence beginning "For South Asia..." - Does this South Asia response occur only with emissions from South Asia, or is it also seen with emissions from elsewhere that transport to South Asia?

*Yes, this could easily be misunderstood. We have changed this to 'For BC emitted in South Asia'.*

Technical issues:

line 14: "a rate of" -> "a factor of"

*This has been changed.*

line 17: "BC emitted in South Asia shows a different geographical pattern by changing..." - This sentence needs clarification. Geographical pattern of which variable(s)?

*We have added: 'geographical pattern in surface temperatures ..'*

line 75: "according different" -> "according to different"

line *This has been changed.*

84: "perturbed by 10 and 20" -> "perturbed by factors of 10 and 20"

*This has been changed.*

line 128: "global BC burden": This is the  global BC burden, correct?

*No, this is the total global BC burden change, as stated in the manuscript, but we have changed the text to be clearer. When we double the emissions, we more than double the burden (117% increase (typo 217 % has been changed, thanks for noting this!). We have also changed the description further down: 'For the simulations with the high emission rate, the relative concentration increase normalized to global emissions is largest at high altitudes above 200 hPa (25-55% increase shown in zonal mean plots of concentrations normalized to emissions per altitude in Fig S3-S6).*

line 132: Please clarify the meaning of "emission-driven vs. concentration-driven BC"

*We agree that this was confusing; we have changed this to: 'BC perturbed by increasing the BC emission rate compared to increasing the BC concentrations.'*

line 144-145: Sentence beginning "For the Arctic..." is unclear. Please rework this.

[revised manuscript text omitted]

Samset, B. H., and Myhre, G.: Climate response to externally mixed black carbon as a function of altitude, J. Geophys. Res., 315 120, 2913-2927, 10.1002/2014JD022849, 2015.

[revised manuscript text omitted]

---

## Author Comment (AC2) · 2 Dec 2019

Review of "Surface temperature response to regional black carbon emissions: Do location and magnitude matter?" by Sand et al., submitted to Atmospheric Chemistry and Physics.

*We would like to thank the reviewer for the dedicated time he/she spent on reading the manuscript and providing valuable feedback to us. A point-by-point response is given below.*

This paper discusses the impacts of regional emissions of black carbon on global and regional temperatures. Scenarios are examined with different levels of emissions. One conclusion is that a higher emission rate results in lower temperature change per unit emissions, except in the Arctic. Impacts of emissions from North America and Europe are found to be similar to each other but different from emissions from East Asia. Overall, the paper contributes to our understanding of the impacts of black carbon. However, the paper can be improved if the model is described more completely and if previous understanding of black carbon climate effects were discussed more completely. Also, the authors do not discuss the physical processes they are not including that could affect results. Finally, there is no evaluation of the model. Below are more specific comments.

With respect to the model, there should be some validation of baseline cloud fields and aerosol optical depth and aerosol absorption optical depth against satellite and/or in situ data. Some validation is required in virtually all papers.

*A comprehensive validation of the same version of the model has already been performed in Bentsen et al (2013) and Kirkevåg et al (2013) and this is the main reason why we chose this model version. We have added the following: 'A comprehensive assessment of the mean model state is available in Bentsen et al. (2013). Here it is shown that the global mean energy fluxes and associated cloud forcing are close to or within the observations range, but that the cloud cover is underestimated. The aerosols fields are evaluated in Kirkevåg et al. (2013). Here it is shown that global BC concentrations is underestimated by -18 % compared to observations. The global anthropogenic aerosol DRF is -0.08 $Wm^{-2}$ and the indirect radiative forcing is -1.2 $Wm^{-2}$. '*

Abstract. "BC emissions are increase by a rate of 10 and 20. . ." 10 and 20 what? A factor of 10 and 20? 10 and 20%? Please clarify.

*We have changed 'rate' to 'factor'.*

Introduction. "There has been a growing interest for reducing black carbon emissions to slow global warming emissions and improve air quality." This was proposed at least as far back as 2002 in Jacobson (2002), which states in the abstract, "Reducing BC + OM will not only slow global warming but also improve human health." Please include this information.

*We have added this reference.*

Introduction. "BC absorbs solar radiation and therefore rapidly influence heating rates, humidity and clouds in the atmosphere." That feedback, along with 11 other feedbacks of black carbon to climate, is discussed Jacobson (2002), Sections 3.1-3.12. Please discuss.

*We agree that BC climate effects were not described in detail, and we have expanded the Introduction to include more references and examples of responses and feedbacks.*

NorESM. Please state explicitly how, in your model, "aerosols can act as CCN based on their size and composition." Is it an empirical parameterization? Do cloud drops of different size and composition grow physically in time, or do you assume an equilibrium cloud drop size distribution.

*There is not an empirical parameterization of cloud droplets in NorESM. CDNC and LWC are prognostically calculated, with CCN activation estimated based on supersaturations calculated from Köhler theory (Abdul-Razzak and Ghan, 2000) using a sub-grid scale vertical velocity. The rate of activation depends on the size distribution and the hygroscopicity. The cloud droplet size distribution is affected by microphysical processes and varies between the time steps. We agree that the model description was too brief. We have added the information above to Methods.*

Please state explicitly in the text whether you include Cloud Absorption Effects I and II of black carbon (Jacobson, 2012). CAE I is cloud burnoff due to absorption by black carbon inclusions within individual cloud drops and CAE II is cloud burnoff due the hydration of BC-containing aerosol particles between cloud drop at the ~100% relative humidity of the cloud, and the resulting enhanced absorption and heating of the cloud due to optical focusing. If you do not include it, please discuss briefly the potential impacts of not including this treatment.

*The CAE I effect is, as in most global climate models, not included in NorESM. We have included in the Methods: 'Any cloud 'burn-off'-effects due to absorption by BC within individual cloud droplets are not included'.*

*Studies show that the lensing effect might enhance BC absorption, but the magnitude of this enhancement remains controversial (Lack et al., 2009; Cappa et al., 2012) due to the various assumptions (coating thickness/morphology/mixing geometry) (e.g. Adachi et al., 2010, Scarnato et al., 2013). Coating of BC will also reduce the BC atmospheric lifetime (Boucher et al. 2016). It is therefore difficult to discuss this in a profound way, unfortunately.*

*Adachi, K., Chung, S. H., and Buseck, P. R.: Shapes of soot aerosol particles and implications for their effects on climate, J. Geophys. Res.-Atmos., 115, D15206, https://doi.org/10.1029/2009JD012868, 2010*

*Boucher, O., Balkanski, Y., Hodnebrog, Ø., Myhre, C. L., Myhre, G., Quaas, J., Samset, B. H., Schutgens, N., Stier, P., and Wang, R.: Jury is still out on the radiative forcing by black carbon, Proceedings of the National Academy of Sciences, 113, E5092-E5093, 10.1073/pnas.1607005113, 2016.*

*Scarnato, B. V., Vahidinia, S., Richard, D. T., and Kirchstetter, T. W.: Effects of internal mixing and aggregate morphology on optical properties of black carbon using a discrete dipole approximation model, Atmos. Chem. Phys., 13, 5089–5101, https://doi.org/10.5194/acp-13-5089-2013, 2013.*

*Lack, D. A., Cappa, C. D., Cross, E. S., Massoli, P., Ahern, A. T., Davidovits, P., and Onasch, T. B.: Absorption Enhancement of Coated Absorbing Aerosols: Validation of the Photo-Acoustic Technique for Measuring the Enhancement, Aerosol Sci. Tech., 43, 1006–1012, https://doi.org/10.1080/02786820903117932, 2009.*

*Cappa, C. D., Onasch, T. B., Massoli, P., Worsnop, D. R., Bates, T. S., Cross, E. S., Davidovits, P., Hakala, J., Hayden, K. L., Jobson, B. T., Kolesar, K. R., Lack, D. A., Lerner, B. M., Li, S.-M., Mellon, D., Nuaaman, I., Olfert, J. S., Petäjä, T., Quinn, P. K., Song, C., Subramanian, R., Williams, E. J., and Zaveri, R. A.: Radiative Absorption Enhancements Due to the Mixing State of Atmospheric Black Carbon, Science, 337, 1078–1081, https://doi.org/10.1126/science.1223447, 2012.*

Please state explicitly how "The BC on snow effect is included in the model." Do you model the size-resolved deposition of BC (through wet and dry deposition) to the surface. How is radiative transfer solved through the snow if it is? If it is not, please state so explicitly.

*We have changed this to: Deposited aerosols on snow-cover and bare sea-ice are calculated prognostically and include hydrophobic and hydrophilic black carbon and dust. While the effects of deposition of these light-absorbing aerosols are taken into account, we have not calculated the radiative transfer of BC in snow explicitly.*

What is the mixing state of BC aerosol components in the model? Externally mixed, well-internally mixed, or something in between?

*We have added the following: 'BC primary particles are assumed externally-mixed. BC is internally-mixed with organic matter and $SO_4$, the latter through either condensation or coagulation. The mass of internally mixed coating species is used in the calculations of the optical properties and cloud droplet number concentration, but it does not alter wet deposition rates'*

Abstract. "For these regional BC emissions perturbations, we find that the effective radiative forcing is not a good measure of the climate response." Although it is not clear if your model includes this, one explanation may be that "when absorbing aerosols exist in clouds, instantaneous direct radiative forcing (DRF) and surface temperature change are anticorrelated because when absorbing aerosol burns off a cloud, the aerosol DRF decreases due to a decrease in optical focusing, yet surface temperature escalates rapidly due to the pouring in of sunlight to the surface." (Jacobson, 2014). Please discuss.

*Again, as this effect is not included in NorESM unfortunately, is it difficult to discuss this.*

Figure 5. Change figure captions to "Delta TOA" rather than just "TOA" Are these "net downward minus upward" irradiances with versus without black carbon emissions? If so, please clarify.

*We have changed the caption to: 'Change in delta TOA net SW and LW radiative fluxes (sum of downward and upward) for the 10×BC in East Asia perturbation minus the baseline.'*

The text mentions that the Indian Monsoon was displaced. How was it displaced? In what direction and what was the magnitude in wind speed change? Were Monsoon wind speeds decreased due to black carbon? Were globally or regionally averaged wind speeds decreased as might be expected (Jacobson and Kaufman, 2006).

*We have expanded this section, and added more references:*

*'This cooling is linked to a strong decrease in downwelling SW radiation over India caused by the absorption and scattering by BC aerosols, less upward motion and an increase in low level clouds. Northern hemisphere BC emissions cause a northward displacement of the ITCZ that is most pronounced for emissions in South Asia and a corresponding shift in the Indian monsoon. The tropospheric jet is moving northward to adjust for a northward shift in the meridional temperature gradient (not shown).This type of displacement caused by a hemispherical asymmetric heating is a robust feature found in several climate model studies (Chung and Seinfeld, 2005; Jones et al., 2007; Meehl et al., 2008; Allen et al., 2012, Voigt et al. 2017; Kovilakam and Mahajan, 2015)'*

**Surface temperature change [K]**

[Figure]

**Figure 1: Surface temperature change (in K) for the perturbed runs minus the baseline. The emission region is given on each panel (from top: Europe, North America, South Asia, East Asia). In the right-side column the emission rate is doubled compared to the left-side column. The stippled areas represent statistically significant changes (p<0.05).**

ΔT per global emissions [K/Tgyr$^{-1}$]

[Figure]

**Figure 2: Regional surface temperature response per BC emissions (K/Tgyr$^{-1}$). The emission location (Europe, North America, South Asia, East Asia) is given on top of each plot. The response region (latitude band) is given on the x-axis. The emission rate is doubled on the right-side bars (darker colours) compared to the left-side bars (lighter colours). The error bars represent the standard error of mean.**

360

Burden per global emissions [Tg/Tgyr$^{-1}$]

[Figure]

Figure 3: Regional BC burden change per global BC emissions (Tg/Tgyr$^{-1}$). The emission location (Europe, North America, South Asia, East Asia) is given on top of each plot. The response region (latitude band) is given on the x-axis. The emission rate is doubled on the right-side bars (darker colours) compared to the left-side bars (lighter colours). The error bars represent the standard error of mean.

365

[Figure]

**Figure 4: Regional surface temperature response per global BC (a) DRF and (b) ERF (in K/Wm⁻²). The emission location (Europe, North America, South Asia, East Asia) is given on top of each plot. The response region (latitude band) is given on the x-axis. The emission rate is doubled on the right-side bars (darker colours) compared to the left-side bars (lighter colours). The error bars represent the standard error of mean.**

370

[Figure]

**Figure 5: Change in delta TOA net SW and LW radiative fluxes ( sum of downward and upward) for the 10×BC in East Asia perturbation minus the baseline. Top row shows the runs with fixed SSTs, and bottom row shows the runs with fully coupled ocean. The stippled areas represent statistically significant changes (p<0.05).**

[Figure]

**Figure 6: Response in high clouds (located above 400 hPa) (a) and low clouds (surface up to 700 hPa) (b) in (%). The emission location (Europe, North America, South Asia, East Asia) is given on top of each plot. In the right-side column the emission rate is doubled compared to the left-side column.**

[Figure]

**Figure 7: A comparison between ARTP calculated and NorESM calculated regional surface temperature response per unit BC emissions (K/Tg/yr⁻¹). The lowest emission perturbation is shown here (10×Europe, 10×North America, 10×South Asia, 5×East Asia). The response region (latitude band) is given on the x-axis. The error bars for NorESM represent the standard error of mean.**

**Tables**

**Table 1: Overview of the model simulations in this study and the global BC emissions in each simulation (in Tg BC/yr).**

| | | |
|---|---|---|
| Control | Year 2000 control run 3×100 years fully coupled equilibrium simulations | 7.7 |
| 10 × Europe | Same as control, but with BC emissions in Europe multiplied by 10 | 12,8 |
| 20 × Europe | Same as control, but with BC emissions in Europe multiplied by 20 | 17,9 |
| 10 × North America | Same as control, but with BC emissions in North America multiplied by 10 | 11,9 |
| 20 × North America | Same as control, but with BC emissions in North America multiplied by 20 | 16,1 |
| 10 × South Asia | Same as control, but with BC emissions in South Asia multiplied by 10 | 13,5 |
| 20 × South Asia | Same as control, but with BC emissions in South Asia multiplied by 20 | 19,3 |
| 5 × East Asia | Same as control, but with BC emissions in East Asia multiplied by 5 | 15,0 |
| 10 × East Asia | Same as control, but with BC emissions in East Asia multiplied by 10 | 22,4 |

**Table 2: Surface temperature change per global BC emissions (in K/Tg yr⁻¹). Bold numbers represent statistically significant differences between the emission rates (using 10 vs. 20, or 5 vs. 10 for East Asia).**

|  | Arctic | Mid lats | Tropics | SH |
|---|---|---|---|---|
| 10 × Europe | **0,10** | **0,08** | 0,02 | **0,01** |
| 20 × Europe | **0,06** | **0,06** | 0,01 | **0,00** |
| 10 × North America | 0,10 | 0,06 | 0,01 | 0,01 |
| 20 × North America | 0,08 | 0,06 | 0,01 | 0,00 |
| 10 × South Asia | **0,08** | 0,04 | 0,01 | 0,01 |
| 20 × South Asia | **0,04** | 0,04 | 0,01 | 0,01 |
| 5 × East Asia | **0,08** | 0,04 | 0,01 | **0,01** |
| 10 × East Asia | **0,05** | 0,04 | 0,01 | **0,01** |

395

**Table 3: Surface temperature change per global direct radiative forcing (in K/Wm⁻²). Bold numbers represent statistically significant differences between the emission rates (using 10 vs. 20, or 5 vs. 10 for East Asia).**

|  | Arctic | Mid lats | Tropics | SH |
|---|---|---|---|---|
| 10 × Europe | **1,74** | **1,39** | 0,31 | **0,23** |
| 20 × Europe | **1,20** | **1,18** | 0,26 | **0,06** |
| 10 × North America | 1,57 | 1,04 | 0,19 | 0,15 |
| 20 × North America | 1,36 | 0,96 | 0,21 | 0,08 |
| 10 × South Asia | **0,77** | 0,38 | 0,13 | 0,08 |
| 20 × South Asia | **0,39** | 0,39 | 0,11 | 0,08 |
| 5 × East Asia | **1,20** | 0,65 | 0,19 | **0,15** |
| 10 × East Asia | **0,81** | 0,60 | 0,17 | **0,09** |

400

---

## Author Response (AR2)

Editor Decision: Publish subject to minor revisions (review by editor) '

*We are grateful to the editor for the time and constructive comments on our manuscript, and for considering our paper for publication.*

Comments to the Author:

lines 12-13: Please indicated the period over which you performed the reference and sensitivity simulations.

*We have specified this in the abstract.*

lines 100 and 102: Please replace "Here it is shown" by "they have shown" or similar.

*This has been changed.*

abstract (at the end), please indicate the limitations of this modeling study (e.g. not including the cloud burn-off effects or other major simplifications that can introduce uncertainties).

[revised manuscript text omitted]